# Doubly Sparse: Sparse Mixture of Sparse Experts for Efficient Softmax Inference

## Abstract

Computations for the softmax function in neural network models are expensive when the number of output classes is large. This can become a significant issue in both training and inference for such models. In this paper, we present Doubly Sparse Softmax (DS-Softmax), *Sparse Mixture of Sparse Experts*, to improve the efficiency for softmax inference. During training, our method learns a two-level class hierarchy by dividing entire output class space into several partially overlapping experts. Each expert is responsible for a learned subset of the output class space and each output class only belongs to a small number of those experts. During inference, our method quickly locates the most probable expert to compute small-scale softmax. Our method is learning-based and requires no knowledge of the output class partition space a priori. We empirically evaluate our method on several real-world tasks and demonstrate that we can achieve significant computation reductions without loss of performance.

## 1 Introduction

Deep learning models have demonstrated impressive performance in many classification problems (LeCun et al., 2015). In many of these models, the softmax function/layer is commonly used to produce categorical distributions over the output space. Due to its linear complexity, the computation for the softmax layer can become a bottleneck with large output dimensions, such as language modeling (Bengio et al., 2003), neural machine translation (Bahdanau et al., 2014) and face recognition (Sun et al., 2014). In some models, softmax contributes to more than 95% computation. This becomes more of an issue when the computational resource is limited, like mobile devices (Howard et al., 2017).

Many methods have been proposed to reduce softmax complexity for both training and inference phases. For training, the goal is to reduce the training time. Sampling based (Gutmann & Hyvärinen, 2012) and hierarchical based methods (Goodman, 2001; Morin & Bengio, 2005) were introduced. D-Softmax (Chen et al., 2015) and Adaptive-Softmax (Grave et al., 2016), construct two level-hierarchies for the output classes based on the unbalanced word distribution for training speedup. The hierarchies used in these methods are either pre-defined or constructed manually, which can be unavailable or sub-optimal. Unlike training, in inference, our goal is not to computing the exact categorical distribution over the whole vocabulary, but rather to search for top-K classes accurately and efficiently. Existing work (Shrivastava & Li, 2014; Shim et al., 2017; Zhang et al., 2018) on this direction focus on designing efficient approximation techniques to find the top-K classes given a *trained* model. Detailed discussions of related works are to be found in Section 4.

Our work aims to improve the inference efficiency of the softmax layer. We propose a novel Doubly Sparse softmax (DS-Softmax) layer. The proposed method is motivated by (Shazeer et al., 2017), and it learns a two-level overlapping hierarchy using *sparse* mixture of *sparse* experts. Each expert is trained to only contain a small subset of entire output class space, while each class is permitted to belong to more than one expert. Given a set of experts and an input vector, the DS-Softmax first selects the top expert that is most related to the input (in contrast to a dense mixture of experts), and then the chosen expert could return a scored list of most probable classes in it sparse subset. This method can reduce the linear complexity in original softmax significantly since it does not need to consider the whole vocabulary.

We conduct experiments in different real tasks, ranging from language modeling to neural machine translation. We demonstrate our method can reduce softmax computation dramatically without loss of prediction performance. For example, we achieved more than 23x speedup in language modeling and 15x speedup in translation with similar performances. Qualitatively, we demonstrate learned two-level overlapping hierarchy is semantically meaningful on natural language modeling tasks.

## 2 DS-SOFTMAX: SPARSE MIXTURE OF SPARSE EXPERTS

### 2.1 BACKGROUND

Before introducing our method, we first provide an overview of the background.

**Hierarchical softmax.**   Hierarchical softmax uses a tree to organize output space where a path represents a class (Morin & Bengio, 2005). There are a few ways to construct such hierarchies. Previous work(Morin & Bengio, 2005; Chen et al., 2015; Grave et al., 2016) focus on building hierarchies with prior knowledge. Other approaches, like Mnih & Hinton (2009), performed clustering on embeddings to construct a hierarchy. Our work aims to learn a two-level hierarchy while the major difference is that we allow overlapping in the learned hierarchy.

**Sparsely-gated mixture-of-experts.**   Shazeer et al. (2017) designed a sparsely gated mixture of experts model so that outrageously large networks can achieve significantly better performance in language modeling and translation. They borrowed conditional computation idea (Bengio et al., 2015) to keep similar computation even though the number of parameters increases dramatically. Their proposed sparsely-gated Mixture of Experts (MoE) only use a few experts selected by the sparsely gating network for computation on each example. The original MoE cannot speedup softmax computation but serves as an inspiration for our model design.

**Group lasso.**   Group lasso has been commonly used to reduce effective features in linear model (Friedman et al., 2010; Meier et al., 2008). Recently, it has been applied in a neural network for regularization (Scardapane et al., 2017) and convolutional deep neural network speedup (Wen et al., 2016). It has been demonstrated as an effective method to reduce the number of nodes in the neural network. In this work, we use group lasso to sparsify the experts.

### 2.2 OUR METHOD

Goodman (2001) studied a two-level hierarchy for language modeling, where each word belongs to a unique cluster. (A "cluster" here refers to a cluster of words.) From this perspective, our method can be thought of as an extension of their method to allow overlapping hierarchy. This is because, in language modeling, it is often difficult to exactly assign a word to a single cluster. For example, if we want to predict the next word of "I want to eat ___" and one possible correct answer is "cookie", we can quickly notice that possible answer belongs to something eatable. If we only search for the right answer inside words with the eatable property, we can dramatically increase the efficiency. Even though words like "cookie" are one of the correct answers, it might also appear under some non-edible context such as "a piece of data" in computer science literature. Thus, a two-level overlapping hierarchy can naturally accommodate word homonyms like these by allowing each word to belong to more than one cluster. We believe that this observation is likely to be true in other applications besides language modeling.

Doubly Sparse softmax (DS-Softmax) is designed to capture such overlapped two-level hierarchy among output classes. In DS-Softmax, the first level is the *sparse mixture* and second level contains several *sparse experts*. (Here an expert can be thought as a similar concept as a cluster.) The sparse mixture chooses the right expert/cluster while sparse experts are responsible to separate the full output space into multiple, overlapped and small class clusters. The design of mixture gating is inspired by Shazeer et al. (2017), where each expert in their model needs to search whole output space. In contrast, DS-Softmax only searches a small subset of the output space and inference becomes significantly faster than full softmax.

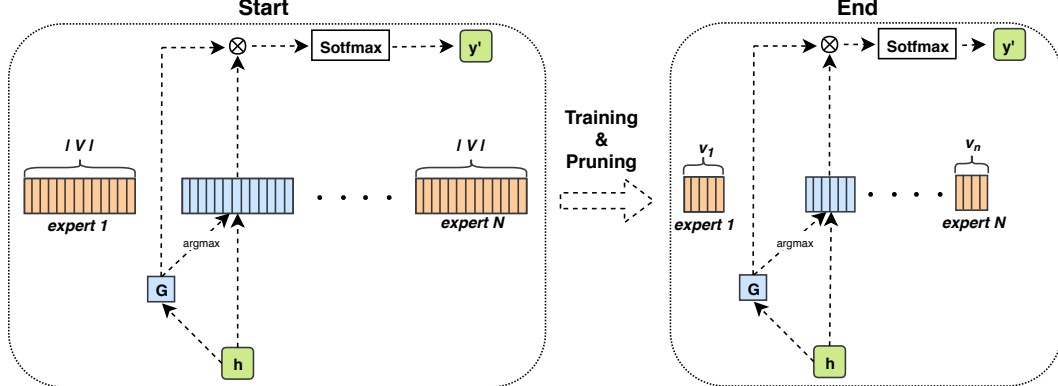

Figure 1: Overview of DS-Softmax. Initial model is similar to sparsly gating mixture of experts model. After pruning, each expert will have partial outputs $v_n$ instead of $|V|$.

**Sparse gating.**  The first level of sparsification is a sparse gating mechanism inspired by Shazeer et al. (2017), which is designed to choose the right experts. The sparse gating outputs a sparse activation over a set of experts. For faster inference purpose, only the top-one expert is chosen. One major difference comparing to Shazeer et al. (2017) is described as follows. Suppose we have $K$ experts . Given input activation vector $h \in \mathbb{R}^d$ and gating network weight $W^g \in \mathbb{R}^{K \times d}$, gating values $G_k(h)$, $k = 1, ..., K$, are calculated and normalized prior to the selection as shown in Eq. 1 and then we choose the gate with the largest value $g_k = \max_i G_i(h)$ and set all other gates to be zero, which means the corresponding $k$-th expert is chosen.

$$G_k(h) = \frac{\exp(W_k^g h)}{\sum_{k'} \exp(W_{k'}^g h)},$$
$$g_k = \begin{cases} G_k(h), & \text{if } k = \arg\max_i G_i(h), \\ 0, & \text{otherwise.} \end{cases} \tag{1}$$

Where $W^g \in \mathbb{R}^{K \times d}$ is the weighting matrix for group selection. We should mention that despite only the top-1 expert is selected, Eq. 1 still allows the gradient to be back-propagated to whole $W^g$. Given the sparse gate, we can compute the probability of class $c$ under the context $h$ as:

$$O(h) = p(c|h) = \frac{\exp(\sum_k g_k W_{(c,k)}^e h)}{\sum_{c'} \exp(\sum_k g_k W_{(c',k)}^e h)}, \tag{2}$$

where $W_{(c,k)}^e \in \mathbb{R}^d$ is softmax embedding weight vector for class $c$ in expert $k$, and only one $g_k$ (the chosen expert) is nonzero in the formulation above.

Note that our design of Eq. 2 is different from Shazeer et al. (2017), in that the gating values $g_k$ are put inside the softmax. This choice is critical as selecting top-1 expert under Shazeer et al. (2017), where normalization is done after top-K experts are selected, results in a constant that carries no gradient information. Also, the gating values can be interpreted as an inverse temperature term for final categorical distribution produced by the chosen expert $k$ Hinton et al. (2015), shown in Eq. 2. A smaller $g_k$ gives a more uniform distribution and larger $g_k$ corresponds to a sharper one.

**Sparse experts with group lasso.**  The second level sparsification is the sparse experts that output a categorical distribution for only a subset output classes. To sparsify each expert, we apply group lasso loss to restrain the $W_{(c,k)}^e$, shown in Eq. 3. Then, pruning is carried out for $W_{(c,k)}^e$ during training with $\gamma$, a lasso threshold according to Eq. 4.

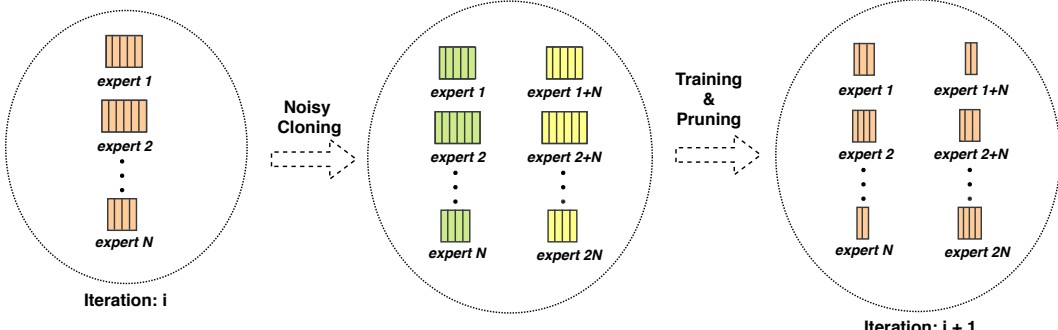

Figure 2: The mitosis training scheme: the sparsity is inherited when parent experts produce offspring, reducing the memory requirements for training with more experts.

$$\mathcal{L}_{lasso} = \sum_k \sum_c \|\hat{W}^e_{(c,k)}\|_2, \tag{3}$$

$$\hat{W}^e_{(c,k)} = \begin{cases} W^e_{(c,k)}, & \text{if } \|W^e_{(c,k)}\|_2 > \gamma, \\ 0, & \text{otherwise.} \end{cases} \tag{4}$$

**Loading Balance.** We denote the sparsity percentage out of full softmax in $k$-th expert as sparsity$_k$ and proportion of $k$-th expert activated as utilization$_k$. The sparsity percentage indicates the percent of pruned output classes. Then, the overall speedup compared to the full softmax can be calculated as $1/\sum_k(\text{utilization}_k * \text{sparsity}_k)$. Thus, better utilization is essential for speedup as well. For example, there is no speedup if the expert with full output space is always chosen. We borrow a similar loading balance function from Shazeer et al. (2017) in Eq. 5. It encourages the utilization percentage of each expert to be balanced by maximizing the coefficient of variation (CV) for gating outputs. In addition, to encourage the exclusiveness of classes, we incorporate group lasso loss on the expert level where each class should only exist in only one expert as shown in Eq. 6.

$$\mathcal{L}_{load} = -\text{CV}\left(\sum_{h \in H(x)} G(h)\right), \tag{5}$$

$$\mathcal{L}_{expert} = \sum_k \sqrt{\sum_c \|W^e_{(c,k)}\|_2^2}. \tag{6}$$

**Mitosis training.** Memory might become a bottleneck during training if we initialize all experts with full softmax. Therefore, we design a training scheme, called mitosis training, to reduce the memory requirement. The method is to initialize with a smaller model (fewer number of experts) and gradually breed to a bigger number after noisy cloning showed in Fig. 2. For each cloning, the sparsity is inherited so that less memory is required. For example, in one of our experiments, we only need 3.25x memory with 64 experts compared to a full softmax implementation.

**The final training algorithm.** Our final training objective, $\mathcal{L}_{all}$, consists of a combination of the related contributions discussed above. We describe our training procedure in Algorithm 1.

## 3 EXPERIMENTS

We evaluate the proposed method on both synthetic and real tasks. For the synthetic task, our goal is to demonstrate that our learning method could discover the hidden two-level hierarchy automatically. In real tasks, we also evaluate both theoretical speedup (reduction in FLOPs) and real device speedup (latency on CPU). Three different real tasks are included: natural language modeling, neural machine translation, and Chinese handwritten character recognition.

---

**Algorithm 1**

---

1: **Initialization:** Let $x$ be the input, $y$ be the corresponding label, $H$ be the pre-trained function, and $D(y', y)$ a distance function, and the hyper-parameter $t$ denotes target performance. Set $W^e \leftarrow$ parameters for experts and $W^g \leftarrow$ parameters for the gating network.
2: **while** training not converged **do**
3:      $\mathcal{L}_{task} = D(O(H(x)), y)$
4:      $\mathcal{L}_{all} = \mathcal{L}_{task} + \lambda_{load}\mathcal{L}_{load} + \lambda_{lasso}\mathcal{L}_{lasso} + \lambda_{expert}\mathcal{L}_{expert}$
5:      $W^e = W^e - \alpha\frac{\partial}{\partial W^e}\mathcal{L}_{all}(x, y; W^e, W^g)$
6:      $W^g = W^g - \alpha\frac{\partial}{\partial W^g}\mathcal{L}_{all}(x, y; W^e, W^g)$
7:      **if** $L_{task} < t$ **then**
8:          **for all** $W^e_{(c,k)} \in W^e$ **do**
9:              $W^e_{(c,k)} = 0$, if $\|W^e_{(c,k)}\|_2 < \gamma$

---

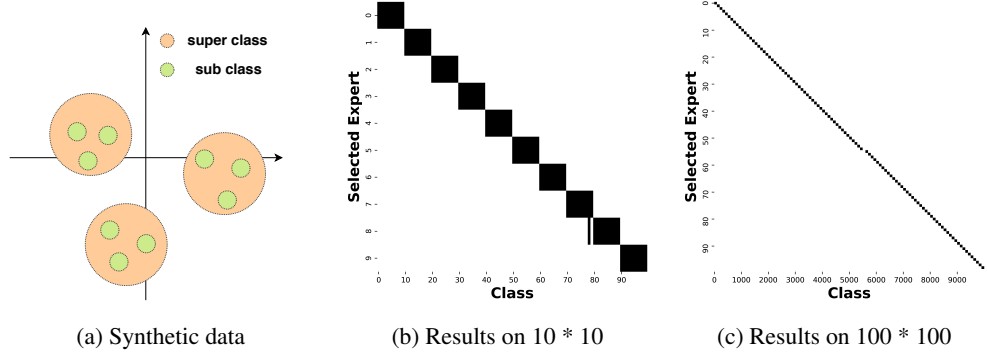

(a) Synthetic data  (b) Results on 10 * 10  (c) Results on 100 * 100

Figure 3: (a) Illustration of data generation. (b) and (c) Results on discovered sparse experts on 10x10 and 100x100 datasets. The x-axis indicates sub class and y-axis shows the selected expert for handling this class. The order of x-axis is arranged through their super class information. For example, each 10 sub classes are belonged to one super class in (b).

We pretrained other layers except DS-Softmax layer for faster converge. For hyper-parameters, $\lambda_{load}$ and threshold are constant. $\lambda_{lasso}$ and $\lambda_{expert}$ share same value but need to be tuned according to performance on a validation set.

### 3.1 SYNTHETIC TASK

One two-level hierarchy synthetic dataset is illustrated Fig 3a. The data generation detail is shown in Appendix A. Two different sizes are evaluated, 10x10 (super class x sub class) and 100x100. The result is illustrated in Fig. 3b and Fig. 3c. We found DS-Softmax can perfectly capture the hierarchy. For sanity check and visualization purposes, the ground-truth two hierarchy in the synthetic data does not have overlapping. We did further analysis on the results on 10x10 synthetic as shown in Fig 4 to study the effects of group lasso and balancing. As we can see, both are important to our model.

### 3.2 LANGUAGE MODELING

Language modeling usually contains a large output dimension. We use two standard datasets for word level language modelling: PennTree Bank (PTB) [Marcus et al. (1994)] and WikiText-2 [Merity et al. (2016)]. The output dimensions are 10,000 and 33,278 respectively. Standard two-layers LSTM model [Gers et al. (1999)] with 200 hidden size is used [1]. Top 1, Top 5 and Top 10 accuracies on testing datasets are used as metric rather than perplexity. Accuracy is a common metric [Chen et al. (1998)] in natural language modeling especially in a real application when the extrinsic reward is

---

[1]https://github.com/tensorflow/models/tree/master/tutorials/rnn/ptb

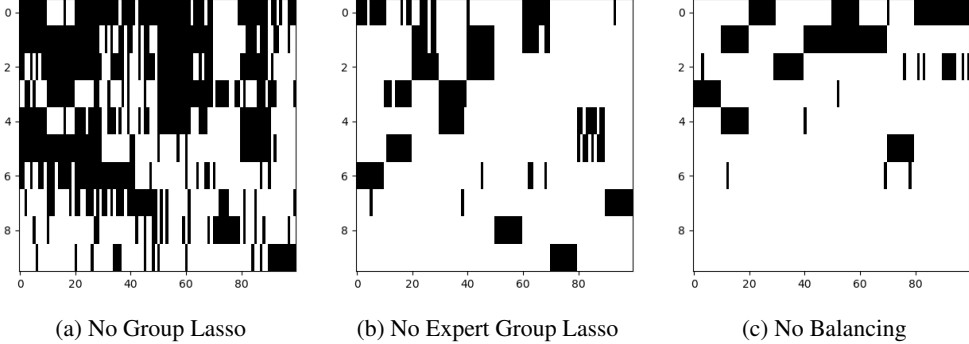

| (a) No Group Lasso | (b) No Expert Group Lasso | (c) No Balancing |

Figure 4: Ablation analysis of each loss component by removing it. (a), (b) and (c) illustrate the model trained without group lasso, expert level group lasso and balancing factor respectively.

| Task | Method | Testing Accuracy | | | FLOPs Reduction |
|------|--------|-------|-------|--------|-----------------|
|      |        | Top 1 | Top 5 | Top 10 |                 |
| PTB (10,000) | Full Softmax | 0.252 | 0.436 | 0.515 | 1x |
|      | DS-8  | 0.257 | 0.448 | 0.530 | 2.84x |
|      | DS-16 | 0.258 | 0.450 | 0.529 | 5.13x |
|      | DS-32 | 0.259 | 0.449 | 0.529 | 9.43x |
|      | DS-64 | 0.258 | 0.450 | 0.529 | 15.99x |
| wiki-2 (33,278) | Full Softmax | 0.257 | 0.456 | 0.533 | 1x |
|      | DS-8  | 0.259 | 0.459 | 0.536 | 3.52x |
|      | DS-16 | 0.264 | 0.469 | 0.547 | 6.58x |
|      | DS-32 | 0.260 | 0.460 | 0.535 | 11.59x |
|      | DS-64 | 0.259 | 0.458 | 0.533 | 23.86x |

Table 1: Word level natural language modelling results on PTB and WikiText-2. The output dimensions are 10,000 and 33,278 respectively. The top1, top5 and top10 accuracies is used as metric.

given, such as voice recognition. We demonstrate that 15.99x and 23.86x times speedup can be achieved with 64 experts without loss of accuracy shown in Table 1.[2]

### 3.3 NEURAL MACHINE TRANSLATION

As language related task, neural machine translation task is usually used for softmax speedup evaluation. We use IWSLT English to Vietnamese dataset (Luong & Manning, 2015) and evaluate performance by BLEU score (Papineni et al., 2002) with greedy searching. The BLEU is assessed on the testing dataset. The vanilla softmax model is seq2seq (Sutskever et al., 2014) and implemented using tensorflow (Abadi et al.) [3]. Dataset preprocessing is same as original implementation and number of output words are 7,709.

### 3.4 CHINESE CHARACTER RECOGNITION

For Chinese handwriting character recognition task, we use the offline and filtered CASIA dataset (Liu et al., 2011). We chose this task because these type of models are likely to be used on resource-limited devices such as mobile phones. CAISA is popular Chinese character recognition dataset with more than four thousand characters. We removed some special characters for better vanilla softmax model performance. Two-thirds of the data is chosen for training and rest for testing. Table 3 shows the results and we can achieve significant speedup on this task too.

---

[2]We require that at least one copy for each word is kept given some expert. Otherwise, we can achieve more than 80x speedup without loss of accuracy at the cost of not modeling some very low-frequency words.

[3]https://github.com/tensorflow/nmt

| Task | Method | Bleu Score | FLOPs Reduction |
|---|---|---|---|
| IWSLT En-Ve (7,709) | Full Softmax | 25.2 | 1x |
| | DS-8 | 25.3 | 4.38x |
| | DS-16 | 25.1 | 6.08x |
| | DS-32 | 25.4 | 10.69x |
| | DS-64 | 25.0 | 15.08x |

Table 2: Speedup comparison for neural machine translation on IWSLT English to Vietnamese and the vocabulary size is 7,709. Bleu score on testing with greedy searching is used as metric.

| Task | Method | Accuracy | FLOPs Reduction |
|---|---|---|---|
| CASIA (3,740) | Full Softmax | 90.6 | 1x |
| | DS-8 | 90.8 | 1.77x |
| | DS-16 | 90.2 | 2.82x |
| | DS-32 | 89.9 | 4.72x |
| | DS-64 | 90.1 | 6.91x |

Table 3: Result on Chinese handwritten character recognition. The accuracy in testing dataset is reported as the performance for each model.

## 3.5 REAL DEVICE COMPARISON

Real device experiments were conducted on a machine with Two Intel(R) Xeon(R) CPU @ 2.20GHz and 16G memory. All tested models are re-implemented using Numpy. Two configurations of SVD-Softmax Shim et al. (2017) are evaluated, SVD-5 and SVD-10. They use top 5% and 10% dimension for final evaluation in their preview window and window width is 16. Indexing and sorting are computationally heavy for SVD-softmax with Numpy implementation. For a fair comparison, we report latency without sorting and indexing for SVD-softmax. However, regards to full softmax and DS-Softmax, full latency is reported. The latency of each sample is shown in table 4. According to the results, our DS-Softmax can achieve not only better theoretic speedup but also less latency.

## 3.6 MITOSIS TRAINING

In this paragraph, we present some details on the mitosis training scheme on PTB language modeling task. The model is initialized with 2 experts, and clone to 4, 8, 16, 32 and 64 experts sequentially. As demonstrated in Appendix B, the model only requires at most 3.25x memory to train DS-64 model and achieve similar performance, significantly smaller than original 64-fold memory. In addition, the final model has less than 1.5 redundancy. This indicates that the memory required for inference is relatively small.

## 3.7 QUALITATIVE ANALYSIS OF SPARSITY

We also investigate how the sparsity changes during training on language modeling task with PTB dataset, DS-64. For the first layer sparsity, we inspect how certain the expert is chosen. As shown in Fig 5a, the certainty increases after longer training as expected. The percentage of high gating value is used as the indication of certainty, where more high gating value means a higher certainty. For the second layer, we interrogate the left words for each expert manually, and the smallest expert is chosen for human validation. We found the words there are actually semantically related. Also, we found the frequency is highly correlated with redundancy that high frequent words appear in more experts. This is a similar phenomenon as the topic models in (Blei et al., 2003; Wallach, 2006). Detail is shown in Appendix C.

## 4 RELATED WORK

In many use cases of softmax inference, such as language modeling or recommendation systems, only top-K most probable classes are needed. And perhaps to this reason, many existing methods for speeding up softmax inference focus on approximately finding top-K classes given the already trained softmax layer. We term them as post-processing methods. For example, the SVD-Softmax

| Task | Full | | DS-64 (Ours) | | | SVD-5 | | | SVD-10 | | |
|------|------|------|------|------|------|------|------|------|------|------|------|
| Name | Value | ms | Value | FLOPs | ms | Value | FLOPs | ms | Value | FLOPs | ms |
| PTB | 0.252 | 0.73 | **0.258** | **15.99x** | **0.05** | 0.249 | 6.67x | 0.12 | 0.251 | 5.00x | 0.18 |
| Wiki-2 | 0.257 | 3.07 | **0.259** | **23.86x** | **0.12** | 0.253 | 7.35x | 0.43 | 0.255 | 5.38x | 0.63 |
| En-Ve | **25.2** | 1.91 | 25.0 | **15.08x** | **0.12** | 25.0 | 6.77x | 0.39 | 25.1 | 5.06x | 0.42 |
| CASIA | **0.906** | 1.61 | 0.901 | **6.91x** | **0.25** | 0.899 | 3.00x | 0.59 | 0.902 | 2.61x | 0.68 |

Table 4: Comparison with SVD-softmax on real device latency. The "ms" indicates the latency in microseconds. Bold fonts indicate better results.

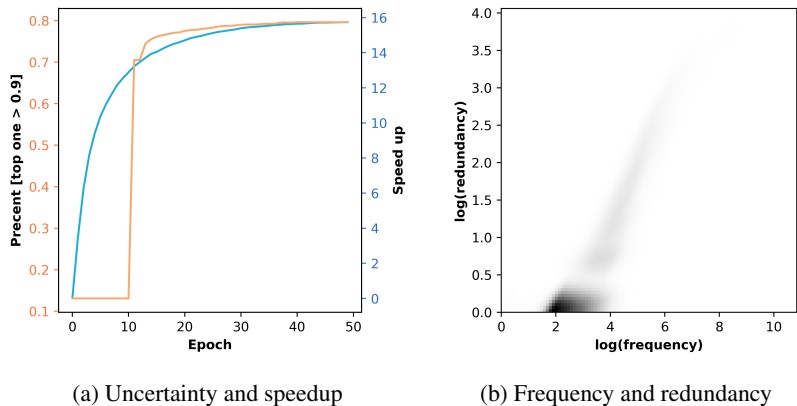

(a) Uncertainty and speedup        (b) Frequency and redundancy

Figure 5: (a) Correlation between uncertainty and training time. We denote uncertainty by the proportion of high gating activation, which is higher than 0.9. (b) Correlation between word frequency and redundancy. Redundancy means the number of appearance for one word in experts. Darker color indicates higher density.

(Shim et al., 2017) method decomposes learned softmax embedding matrix using singular value decomposition and approximates top-K classes through a smaller preview matrix. Approximation methods incorporating traditional searching methods are also popular including Locality Sensitive Hashing (LSH) (Shrivastava & Li, 2014; Maddison et al., 2014; Mussmann et al., 2017; Spring & Shrivastava, 2017) and small word graph (Zhang et al., 2018). Those post-processing methods depend on a well-learned softmax and might become costly when high precision is required.

Other related works proposed to speed up inference by changing the training scheme and we term them as learning-based methods. Our method falls under this category. For example, differentiated softmax (Chen et al., 2015) and adaptive softmax (Grave et al., 2016) can speed up both training and inference by partially activating parameters. Their methods use prior knowledge, such as unbalanced distribution of word frequencies, to obtain a non-overlapping hierarchy for the output class space. The basic ideas were also derived form the original hierarchical methods (Morin & Bengio, 2005; Mnih & Hinton, 2009; Mikolov et al., 2011; Le et al., 2011). Combinations of learning-based and post-processing are also proposed and studied in Levy et al. (2018). Our proposed method learns a two-level hierarchy of experts but each class can belong to more than one experts.

## 5 CONCLUSION

In this paper, we present *doubly sparse: sparse mixture of sparse experts* for efficient softmax inference. Our method is trained end-to-end. It learns a two-level overlapping class hierarchy. Each expert is learned to be only responsible for a small subset of the output class space. During inference, our method first identifies the responsible expert and then perform a small scale softmax computation just for that expert. Our experiments on several real-world tasks have demonstrated the efficacy of our proposed method.

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

## A  SYNTHETIC DATA GENERATION

We design the data generation process as follows: (1) sample super classes $C_i^{\text{super}}$ from given multivariate Gaussian (2) sample sub classes $C_j^{\text{sub}}$ from multivariate Gaussian with mean as $C_i^{\text{super}}$ and a smaller variance (3) The actual input $X_j^{\text{input}}$ is sampled around $C_j^{\text{sub}}$ and even smaller variance, and the corresponding label is $j$. The detail of implementation is shown in Eq. 7.

$$C_i^{\text{super}} \sim \mathcal{N}(0, d^3 I), \quad C_j^{\text{sub}} \sim \mathcal{N}(C_i^{\text{super}}, d^2 I), \quad X_j^{\text{input}} \sim \mathcal{N}(C_j^{\text{sub}}, dI), \tag{7}$$

where we choose $d = 10$ for our experiments.

## B  MITOSIS TRAINING

In mitosis training, the model is first initialized with 2 experts and then is cloned to have to 4, 8, 16, 32 and 64 experts gradually. Each cloning is followed by a regular training with Algorithm 1, which however has different sized $W^e$ and $W^g$. Cloning happens for every 15 epochs and pruning starts 10 epochs after cloning. The result is illustrated in Fig. 6a. As we can see, only less than 3.25x memory is required to train the DS-64 model instead of 64x memory.

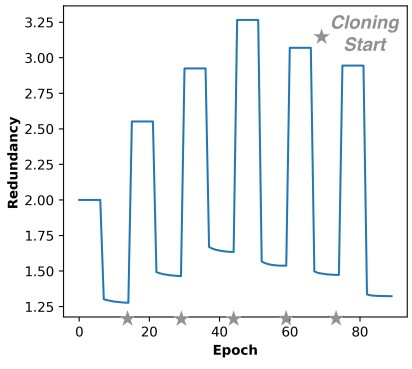
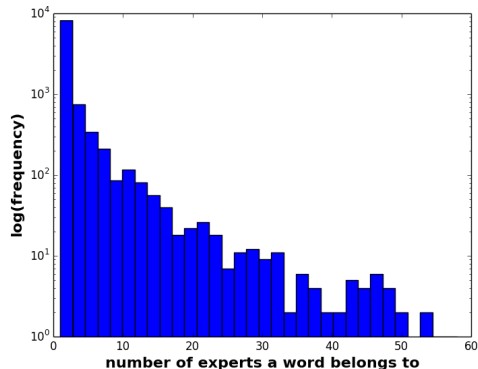

(a) Mitosis Training Result: Illustration of memory requirement needed to train DS-64 starting with DS-2

(b) Overlap Pattern: the x-axis indicates the number of overlap/redundancy. The y-axis indicates the number of words.

## C  ANALYSIS OF EXPERTS

We first demonstrate the redundancy/overlapping pattern in Figure 6b. We choose the smallest expert for qualitative analysis. High-frequency words are filtered out and 64 words remain. We found these words can be classified as three major groups: *money, time* and *comparison* related. For example, "million", "billion", "trillion" appear as money related group. All the weekday words are identified as time-related words. The detail is shown as following:

- **Money related**: million, billion, trillion, earnings, share, rate, stake, bond, cents, bid, cash, fine, payable
- **Time related**: years, while, since, before, early, late, yesterday, annual, currently, monthly, annually, Monday, Tuesday, Wednesday, Thursday, Friday
- **Comparison related**: up, down, under, above, below, next, though, against, during, within, including, range, higher, lower, drop, rise, growth, increase, less, compared, unchanged

