# OpenReview forum: "Doubly Sparse: Sparse Mixture of Sparse Experts for Efficient Softmax Inference"
_ICLR.cc/2019/Conference_

### Official Review · AnonReviewer1 · 2018-11-02
**New method for large scale softmax inference**

**Rating:** 7
**Confidence:** 3

**Review:**

In this paper the authors introduce a new technique for softmax inference. In a multiclass setting, the idea is to take the output of a NN and turn it into a gating function to choose one expert. Then, given the expert, output a particular category. The first level of sparsity comes from the first expert. The second level of sparsity comes from every expert only outputting a limited set of output categories.

The paper is easy to understand but several sections (starting from section 2) could use an english language review (e.g. "search right" -> "search for the right", "predict next word" -> "predict the next word", ...) In section 3, can you be more specific about the gains in training versus inference time? I believe the results all relate to inference but it would be good to get an overview of the impact of training time as well. You motivate some of the work by the fact that the experts have overlapping outputs. Maybe in section 3.7 you can address how often that occurs as well?

Nits:
- it wasn't clear how the sparsity percentage on page 3 was defined?
- can you motivate why you are not using perplexity in section 3.2?

---

> ### Author Response · Authors · 2018-11-14
> **Thank you for your feedback!**
>
> Dear Reviewer:
>
> Thank you for your valuable comments. We have addressed typos in the revision accordingly.  And please find our response as follows.
>
> -  Can you be more specific about the gains in training versus inference time?
>
> We would like to emphasize that the our goal is to speed up the inference time for softmax, so we do not include any comparisons in terms of training time. According to our experiments, most speedup can be achieved in few epochs (given all other layers are pre-trained) so that the training time increase is not significant compared to the original one.
>
> - You motivate some of the work by the fact that the experts have overlapping outputs. Maybe in section 3.7 you can address how often that occurs as well?
>
> Thanks for the suggestion. We demonstrate that ambiguous words are often overlapped between clusters as illustrated in Figure 3(b). We added one more Figure in Appendix B, Figure (b), to demonstrate the distribution of overlapping.
>
> - It wasn't clear how the sparsity percentage on page 3 was defined?
>
> Sorry for the possible confusion. The sparsity in page 3 means the percentage of pruned words. We have added more clarifications in the revised version.
>
> - Can you motivate why you are not using perplexity in section 3.2?
>
> We use top-k accuracy, instead of perplexity, because approximating top-k is required for most inference tasks in practice (see [1]). Perplexity captures the normalized log-likelihood of all possible words, while top-k accuracy is a better measure for inference speedup for top-k retrieval. For example, in some extreme cases, if a word only has a very small probability which makes it unpredictable at all (i.e. couldn’t be retrieved by top-k for any reasonably small k), it could still have a huge impact in terms of perplexity, but has a much smaller impact on top-k accuracy, which seems more reasonable given the goal of top-k retrieval.
>
> [1] Asymmetric LSH (ALSH) for Sublinear Time Maximum Inner Product Search (MIPS), NIPS 2014

---

### Official Review · AnonReviewer3 · 2018-11-03
**Need to discuss more about how Doubly Sparse is superior to Sparsely-Gated MoE**

**Rating:** 4
**Confidence:** 3

**Review:**

The paper proposes doubly sparse, which is a sparse mixture of sparse experts and learns a two-level class hierarchy, for efficient softmax inference.

[+] It reduces computational cost compared to full softmax.
[+] Ablation study is done for group lasso, expert lasso and load balancing, which help understand the effect of different components of the proposed
[-] It seems to me the motivation is similar to that of Sparsely-Gated MoE (Shazeer et al. 2017), but it is not clear how the proposed two-hierarchy method is superior to the Sparsely-Gated MoE. It would be helpful the paper discuss more about this. Besides, in evaluation, the paper only compares Doubly Sparse with full softmax. Why not compare with Sparsely-Gated MoE?

Overall, I think this paper is below the borderline of acceptance due to insufficient comparison with Sparsely-Gated MoE.

---

> ### Author Response · Authors · 2018-11-11
> **Clarifications: Sparsely-Gated MoE (Shazeer et al. 2017) cannot speed-up softmax inference**
>
> Dear reviewer:
>
> We appreciate your comments but it appears that there is some misunderstanding regarding our contribution in this work.
>
> Our work is for softmax inference speedup while Sparse-Gated MoE (MoE) was not designed to do so. It was designed to increase the model expressiveness. It cannot achieve speedup because each expert still contains full softmax space as we mentioned in the background section (page 2 line 21st) and method section (page 2 last 4th line). And since it is slower than the standard softmax by definition, we chose not to compare with it in the paper.
>
> Our algorithm addresses speed up in softmax inference. This is fundamentally different from Sparse-gated MoE. We divide the output space into multiple overlapped subsets. To find top-k predictions, we only search a few subsets. While in full softmax or MoE, the complexity is linear with output dimension. Therefore, we did not include a comparison with Sparsely-Gated MoE in our article and only compare with full softmax.
>
> Just for additional reference, we tested Sparsely-Gated MoE with different experts in PTB dataset; we compared the results to DS-Softmax. As expected, the Sparsely-Gated MoE does not achieve speedup in terms of softmax inference.
>
> ______________________________________________
> Method | Top 1 | Top 5 |Top 10| FLOPs|
> DS-8       | 0.257 | 0.448 | 0.530 | 2.84x |
> MoE-8    | 0.258 | 0.448 | 0.530 |  1x      |
> DS-16     | 0.258 | 0.450 | 0.529 | 5.13x |
> MoE-16  | 0.258 | 0.449 | 0.530 | 1x       |
> DS-32     | 0.259 | 0.449 | 0.529 | 9.43x |
> MoE-32  | 0.259 | 0.450 | 0.531 | 1x       |
> DS-64     | 0.258 | 0.450 | 0.529 |15.99x|
> MoE-64  | 0.260 | 0.451 | 0.531 | 1x       |
> ______________________________________________
>
> * FLOPs means FLOPs reduction (i.e. baseline's FLOPs / target method's FLOPs).

---

### Official Review · AnonReviewer2 · 2018-11-03
**Good empirical results, but only one baseline and poor writing.**

**Rating:** 6
**Confidence:** 3

**Review:**

The present paper proposes a fast approximation to the softmax computation when the number of classes is very large. This is typically a bottleneck in deep learning architectures. The approximation is a sparse two-layer mixture of experts.

The paper lacks rigor and the writing is of low quality, both in its clarity and its grammar. See a list of typos below.

An example of lack of mathematical rigor is equation 4 in which the same variable name is used to describe the weights before and after pruning, as if it was computer code instead of an equation. Also pervasive is the use of the asterisk to denote multiplication, again as if it was code and not math.

Algorithm 1 does not include mitosis, which may have an effect on the resulting approximation.

How are the lambda and threshold parameters tuned? The authors mention a validation set, are they just exhaustively explored on a 3D grid on the validation set?

The results only compare with Shim et al. Why only this method? Why would it be expected to be faster than all the other alternatives? Wouldn't similar alternatives like the sparsely gated MoE, D-softmax and adaptive-softmax have chances of being faster?

The column "FLOPS" in the result seems to measure the speedup, whereas the actual FLOPS should be less when the speed increases. Also, a "1x" label seems to be missing in for the full softmax, so that the reference is clearly specified.

All in all, the results show that the proposed method provides a significant speedup with respect to Shim et al., but it lacks comparison with other methods in the literature.

A brief list of typos:

"Sparse Mixture of Sparse of Sparse Experts"
"if we only search right answer"
"it might also like appear"
"which is to design to choose the right"
sparsly
"will only consists partial"
"with γ is a lasso threshold"
"an arbitrarily distance function"
"each 10 sub classes are belonged to one"
"is also needed to tune to achieve"

---

> ### Author Response · Authors · 2018-11-14
> **Our work focuses on inference speedup, and compares to the best approach (NIPS'17) we were aware of**
>
> Dear Reviewer,
>
> Thank you for your valuable comments. We have revised our writing in the revision, and will further improve its clarity. Please find our response as follows.
>
> - Algorithm 1 does not include mitosis, which may have an effect on the resulting approximation.
>
> Mitosis training can be considered as executing Algorithm 1 for multiple times with an increasing number of experts and inherited initialization from last round by changing W^e and W^g. Also, training with mitosis achieves similar performance as training without it shown in Appendix B, Figure (a).
>
> - How are the lambda and threshold parameters tuned? The authors mention a validation set, are they just exhaustively explored on a 3D grid on the validation set?
>
> The hyper-parameters related to DS-softmax (such as lambda) are tuned according to the performance on a validation dataset. Also, as we mentioned in the paper, only one hyper-parameter (group lasso lambda) needs to be tuned. The heuristic we use to tune group lasso lambda is to increase lambda, starting from a small value, until it hurts the performance. Also threshold and balancing lambda variables are kept fixed as (0.01 and 10).
>
> - Why would it be expected to be faster than all the other alternatives? Wouldn't similar alternatives like the sparsely gated MoE, D-softmax and adaptive-softmax have chances of being faster?
>
> In terms of baselines, SVD-softmax (NIPS’17) was chosen since it is a recent method that provides a significant inference speedup for softmax. Other alternatives, such as D-softmax and adaptive-softmax, focus on training instead of inference speedup. Furthermore, as claimed in their papers, they achieve limited speedup (around 5x) in language modeling, which is much worse than ours. With regards to Sparsely Gated MoE, it cannot speed up inference, since they select expert with full softmax.
>
> We would like to emphasize that most existing methods for inference speedup focus on approximating trained softmax layer, which usually suffers a loss on performance. Our model allows the adaptive adjustment of the softmax layer, achieves speedup through capturing the two-level overlapped hierarchy during training, which is novel and does not suffer from the performance loss.

---

### Author Response · Authors · 2018-11-16
**Summary of the revision and key points**

We thank reviewers for their time and valuable comments. We have revised our article based on reviewers' suggestions.
We want to summarize the key points of this work as follows:

* Our work focuses on speeding up softmax inference given large output dimension and achieved good empirical results on both synthetic and real dataset. For top-k language modeling task on Wiki-2, we can achieve more than 23x without any loss of performance.

* Our method is novel in terms of constructing the two-level overlapping hierarchy of output classes. The hierarchy is captured through the mixture model and group lasso technique. The inference speedup is achieved by such a hierarchy.

* The key difference between our work and existing methods is that our speedup is achieved by learning a new output embedding while most existing methods relied on approximating the trained/fixed embedding. This means our method is orthogonal with them in principle. One key advantage of our method is speedup without any loss while approximation based methods usually suffer the loss of performance.

---

### Meta-Review · Area_Chair1 · 2018-12-13

**Confidence:** 4
**Recommendation:** Reject

**Metareview:**

This work proposes a new approximation method for softmax layers with large number of classes. The idea is to use a sparse two-layer mixture of experts. This approach successfully reduces the computation requires on the PTB and Wiki-2 datasets which have up to 32k classes. However, the reviewers argue that the work lacks relevant baselines such as D-softmax and adaptive-softmax. The authors argue that they focus on training and not inference and should do worse, but this should be substantiated in the paper by actual experimental results.